# Kinematic Studies of the Go/No-Go Task as a Dynamic Sensorimotor Inhibition Task for Assessment of Motor and Executive Function in Stroke Patients: An Exploratory Study in a Neurotypical Sample

**DOI:** 10.3390/brainsci12111581

**Published:** 2022-11-19

**Authors:** Gemma Lamp, Rosa Maria Sola Molina, Laila Hugrass, Russell Beaton, David Crewther, Sheila Gillard Crewther

**Affiliations:** 1School of Psychology and Public Health, La Trobe University, Bundoora, VIC 3086, Australia; 2Centre for Human Psychopharmacology, Swinburne University of Technology, Hawthorn, VIC 3022, Australia

**Keywords:** kinematics, Go/No-Go, inhibition, error processing, reaction time

## Abstract

Inhibition of reaching and grasping actions as an element of cognitive control and executive function is a vital component of sensorimotor behaviour that is often impaired in patients who have lost sensorimotor function following a stroke. To date, there are few kinematic studies detailing the fine spatial and temporal upper limb movements associated with the millisecond temporal trajectory of correct and incorrect responses to visually driven Go/No-Go reaching and grasping tasks. Therefore, we aimed to refine the behavioural measurement of correct and incorrect inhibitory motor responses in a Go/No-Go task for future quantification and personalized rehabilitation in older populations and those with acquired motor disorders, such as stroke. An exploratory study mapping the kinematic profiles of hand movements in neurotypical participants utilizing such a task was conducted using high-speed biological motion capture cameras, revealing both within and between subject differences in a sample of healthy participants. These kinematic profiles and differences are discussed in the context of better assessment of sensorimotor function impairment in stroke survivors.

## 1. Introduction

Upper limb somatosensory loss post stroke is highly prevalent. Such somatosensory loss also limits motor rehabilitation and recovery of function [1] while highlighting the need for understanding the kinematics of motor control to better inform rehabilitation [2]. As of 2020 [3], numerous studies and techniques focussing on deficits of reach to grasp (RtG) co-ordination [4,5] are available, although random clinical trials are far from unanimous regarding ‘value for money’ of some current robotic tools for future rehabilitation [3,6]. Furthermore, an important and often overlooked component of motor function, which has also been shown to be impacted in stroke survivors who experience phenomena such as alien hand syndrome, is inhibition of inappropriate motor impulses [7]. Research into the brain networks underpinning such behaviours is now underway [6,8].

While motor control is usually considered from the viewpoint of muscle control, an alternative approach to such understanding has been recently advanced by Baak, Bock [4], who emphasized findings that stroke-related deficits in ‘sensory motor smoothness’ mainly involve temporal organization of movements rather than muscle control. Previous research has long shown that smoothness occurs during simple movement tasks with low attentional demand and no corrections required [9,10]. However, when the participant is required to make corrections to their motor plan, the new trajectory has been described as “non-smooth”, involving spikes and deviations in the initial trajectory [11,12,13,14]. Most recently, Benedetti, Gavazzi [15] demonstrated that using a simple key press to measure responses in Go/No-Go tasks is limited in only allowing a binary correct/incorrect response, failing to characterise more dynamic aspects on inhibitory control. Hence, Benedetti, Gavazzi [15] used a high-speed gaming mouse to interact with a target on screen and thus were able to characterise behavioural patterns of the cursor trajectory during No-Go trials, which revealed a non-smooth trajectory prior to inhibition of the response. These spikes and deviations in the trajectory were suggested to reflect a period of proactive control (see Figure 1 for processes included in proactive control).

Interestingly, such access to dynamic technology has also allowed us to show fine motor control contrary to expectation in patients following acute ischemic first episode stroke (AIS). We have recorded preferred use of the hand contralateral to lesion hemisphere in 29/31 short-term hospitalized mild–medium AIS patients with left hemisphere lesions who chose to use their right hand to trace and complete three complex visuomotor tasks [16] on an iPad screen and who only made a similar number of motor errors to age matched controls while taking significantly longer, especially when correcting the displacement errors [17]. In both studies [4,17], conscious attention and motor planning seem to be more affected than actual automatic motor control. Hence, we have taken this issue further and examined temporal aspects of conscious control of reaching and grasping actions by young adults in a Go/No-Go task as an exploratory task to assess elements of cognitive control and executive function, which can eventually be applied as an assessment task for post-stroke motor control given that the Go/No-Go literature is accepted as a contentious area in human neuroscience [18].

Many separate investigations segregating the neural localization of the components of stimulus selection and response inhibition utilize brain imaging, commonly fMRI [19,20,21] or EEG [22,23]. However, while the behavioural measures corresponding to the neurological data vary, to date, there are comparatively few kinematic studies detailing the fine spatial and temporal upper limb hand movements associated with the millisecond temporal trajectory of correct and incorrect responses to visually driven Go/No-Go reaching and grasping tasks. Therefore, we aimed to refine the behavioural measurement of correct and incorrect inhibitory motor responses in a Go/No-Go task, with the eventual aim being to better inform stroke rehabilitation research regarding upper limb somatosensory function. Second, we chose to measure other demographic factors associated with psychological/affective state that have been shown to impact visuomotor inhibition in a neurotypical population [24,25] as a means of providing further insight into variables that could impact sensorimotor rehabilitation post stroke given the high risk of depression among older stroke patients [26,27,28].

The Go/No-Go task is commonly considered to involve three stages following stimulus presentation [29]: (i) stimulus detection, (ii) stimulus discrimination and executive decision as to choice of appropriate response and (iii) motor response execution or inhibition. Thus, the preparation of responses in a Go/No-Go task has been theorized to involve two stages of cognitive control, the first occurring proactively and the second reactively [30]. 

Isolating the proactive component is a relatively simple process regardless of whether errors are made as one can simply compare the attentive period prior to stimulus presentation to the period after stimulus presentation and the response (to Go or Not to Go) that has occurred. This can be demonstrated behaviourally with longer RTs shown for Go/No-Go tasks as compared to other control tasks, suggesting that greater conscious executive decision preparation is required [31]. This is an ideal paradigm for fMRI studies and has allowed for much research delineating the underlying neural processes [32]. The difficulty is found in measuring the reactive period, which is subject to many factors in the Go/No-Go task. The veto response is already programmed as a faster, more automatic error correction than for more conscious inhibition, such as that in the stop-signal reaction time task [30,33]. However, Wijesundera’s visuomotor task [17] indicates that it is replanning of the vetoed error motor response that is difficult for acute ischaemic stroke patients. Go/No-Go responses have traditionally been categorized simply as “correct” or “incorrect”, largely relying on reaction time (RT) and number of errors made rather than being measures of temporal agility of executive decision-making, as is necessary for safe motor rehabilitation.

The number of errors made by control subjects is commonly reported to be quite low and subject to task design [34], with participant attention throughout the task [35,36], the influence of preceding targets [37] and many post-error factors, such as learning effects [38], affecting the error number and RT engendered. Indeed, the RT reported in Go/No-Go tasks appears to be impacted by the frequency of No-Go targets, with fewer No-Go targets interrupting, hence resulting in a shorter average RT to “Go” targets [34]. When a participant produces an error response (i.e., reacting to a “No-Go” trial instead of inhibiting this action), the RTs of these responses have commonly been shown to be shorter than correct responses to Go targets by up to 100 ms [35,39] and then longer for the next trials. However, while the number of errors is commonly reported [40,41], examination of these errors has rarely been rigorously considered.

A review conducted in 2013 emphasized the importance of understanding reaching to grasp objects in both healthy and clinical patients in order to understand visuomotor control planning and functioning in the brain and to better inform methodologies aimed at motor rehabilitation [42]. Indeed, the level of visually guided motor control needed is a key quantifier of sensorimotor impairment and prognosis for rehabilitation, and, as such, measuring this coordination accurately is imperative in clinical research associated with rehabilitation following acquired brain injury [43]. Motor RT is known to slow with age regardless of cognitive functioning, so threshold motor RTs can be a confounding factor in studies assessing stroke severity [44]. Thus it is apparent that improving measures of sensorimotor control has important implications for research and translative rehabilitation in clinical populations with acquired brain injury.

While motion capture technology has been established to improve clinical quantification of motor impairments in neurological disorders, with marker-based tracking suggested to be the gold standard [45], to our knowledge, this technology has not been used with motor control studies drawing on increased vigilant attention, such as a Go/No-Go task. Thus, with the prospective aim of understanding the complexity of the neural underpinnings to the cognitive control of impaired motor responses in future research and informing future measures of somatosensory activity in both older controls and stroke patients, we aimed here to refine the temporal trajectory of the kinematic measurement involved in a motor Go/No-Go task while using time-sensitive motion capture cameras (250 Hz) to measure kinematic responses of the hands of young adults as an aid to measurement and interpretation of kinematic recordings of motor deficits in clinical populations [46], enabling temporally superior measures of motor responses [47]. In a sample of neurotypical participants, we predicted that utilization of high-speed biological motor capture, the chronometrically measured timeline of behavioural responses in a Go/No-Go task from RT to task completion, could be refined and enable better delineation of the individual variation in behavioural responses of adults prior to the confounds of aging or neurodegenerative disease. It was further predicted that different types of error response and time to plan and execute correction could also be detected using this technology. Thus, kinematic profiles and neurotypical variation in response possibilities will be described, and the relationship with potential demographic variables, such as anxiety (DASS-21 sub-scores [48]) and attention (AQ sub-scores [49]), will be reported.

## 2. Materials and Methods

### 2.1. Participants

Twenty-eight participants were recruited through social media and La Trobe University. After informed written consent was obtained, participants were screened for hand dominance using the Edinburgh Handedness Scale [50] and for normal corrected vision using the LOGMAR visual acuity chart [51], Ishihara colour vision test [52] and the RANDOT stereopsis test [53]. Demographic information, including age, handedness and laterality quotient [50], were collected. This study was approved by the La Trobe University Human Ethics Committee (UHEC 16-131). At the conclusion of the study, participants were reimbursed for their time with AUD 30 gift vouchers.

### 2.2. Apparatus

A basic RtG visuomotor task [54] based on a seminal study by Castiello, Paulignan [47] was modified to include an increased attentional demand requiring participants to inhibit some reaches. Participants were seated at a desk in a completely darkened room, consistent with the previous study design [47], with a VPixx DATAPixx [55] button box located at a midline position. Three translucent Perspex dowels (2.5 cm diameter and 10 cm height) were placed 35 cm in front of the participants at 15°, 30° and 45° from the midline (see Figure 2a). The dowels were connected to a VPixx system via fibre optic light guides, allowing the dowels to be illuminated with temporal precision according to a programmed task and in response to button presses. Atop the dowels were affixed reflective markers. Similar markers were also attached to the participants’ right index finger, thumb, back of hand, medial wrist and lateral wrist (see Figure 2b). Four Qualisys High Speed Digital Cameras (Oqus) (see Figure 2c) were used to record RtG measurements [56]. The cameras were arranged so that at least two of the cameras captured every movement as a point in the x-y-z space. This allowed 3D recording of each sensor movement at 250 frames per second (250 hz) with millimetre precision [56].

### 2.3. Procedure

Prior to commencing the visuomotor inhibition task, the demographic measures were administered, followed by a simple measure of eye–hand coordination. The demographic measures included a shortened version of the Ravens Progressive Matrices [57] involving 18 matrices designed to measure non-verbal fluid intelligence, a shortened version of the Depression, Anxiety and Stress Scale [48] that comprises 21 questions aimed at measuring relative depression, anxiety and stress as internal states and the Autism Quotient [49] involving questions to assess autistic traits in a neurotypical population. The measure of eye–hand coordination was conducted on an Apple iPad with a rubber-tipped stylus using the SLURP Eye–Hand Coordination Task [16].

Participants were instructed to press the blue button then immediately press and hold the red button, with their fingers in a pincer position, until an object illuminated. This illumination was the cue for the participant to respond; in a Go trial, they were required to reach and pick up the object as quickly and accurately as possible. In a No-Go trial, the participant was required to remain still and continue pressing the red button until the object was no longer illuminated, when they could then initiate the next trial by pressing the blue button. To ensure adequate training for the RtG task, a practice task block of 50 trials was first administered, where only the middle dowel would illuminate. Participants then completed three blocks of the Go/No-Go task. Each block contained 45 trials: 15 illuminations of each left, middle and right dowel in a randomised order. For each block, the participant was instructed to inhibit their reach for a specific dowel and only reach if one of the other two dowels was illuminated. This created 15 No-Go trials and 30 Go trials in each block. Button presses were recorded using electronic switches [55], while biological movement of the dominant hand was captured by the Qualisys cameras [56]. See Figure 3 for timing intervals and possible responses. The order that the blocks were presented to participants was counterbalanced to prevent any learning effects.

Motor RT was measured in two ways: first, using the time of the red button release as measured by the VPixx DataPixx [55], as most studies traditionally report. In second method, RT was defined as the number of milliseconds (ms) *post measurable movement of the first of the hand markers* (either the movement of index finger, back of hand or medial wrist) by greater than 1 mm in the direction of the target object following LO, as measured by the Qualisys cameras [56]. This automatic method chose the first marker to move or the red button release, depending on which occurred first, as the shortest measure of RT.

During the “Go” trials, the following key kinematic events were recorded using the QTM-PE system: peak acceleration (PA; the largest increase in medial wrist velocity from RT to PV, mm/s^2^), peak velocity (PV; highest rate at which the medial wrist marker moved towards the target object from RT to object lift, mm/s), peak deceleration (PD; largest decrease in medial wrist marker velocity from PV to object lift, mm/s^2^) and maximum grip width (MGW; largest distance between index finger and thumb markers between RT and object lift, mm) were all recorded, including the time at which each occurred and the percentage of the movement time in which they occurred. The end of the trial was identified when the z-axis position of the object dowel marker exceeded a 5 mm lift threshold (i.e., object lift) or the trial timed out in the case of a “No-Go” trial (randomised to approximately 2.5 s).

During the “No-Go” trials, responses were broken down into three potential responses:“No Reach” (NR): no button release was recorded, this meant a successful inhibition of reach had occurred“Wrist Movement” (WM): while no button release was recorded, a spike in movement of the medial wrist marker was recorded after LO exceeding a threshold of 2 mm, distinct from other background movement noise. This indicated a partial inhibition or an “almost” instinctual reach to the target despite the instruction to inhibit movement. The number of times these WMs occurred during a block of “No-Go” trials was recorded in addition to the relative RT (number of ms from LO to movement onset) and time of movement stop (ms from LO to movement cessation)“Miss-Reach” (MR): when the participant did release the button and move towards the “No-Go” object but stopped before interacting with the object. The number of times this occurred was recorded in addition to the RT (in either index finger, medial wrist or back of hand), time to peak velocity (ms from RT to PV) and time to movement stop (ms from RT to movement cessation)

The order of the blocks presented was counterbalanced across participants to minimise learning effects. Following the completion of each block, participants were asked to rate their concentration, arousal and performance using a 7-point Likert scale.

### 2.4. Statistical Analysis

Qualisys Track Manager (QTM; Version 2018.1, build 4300) software was utilized to create a model of the hand and dowels, allowing automatic labelling of each marker and manually checking for each participant and task. Once movement data were extracted from QTM [56], the timeline of task events was imported from the VPixx data files and aligned with the QTM data before importing into a custom processing pipeline created in LabVIEW [58] to filter the noise (2nd order, low pass Butterworth filter with a cut-off of 0.125 Hz) while extracting key kinematic events [54]. This processing automatically identified the first marker to move more than 5 mm in each trial in response to the object light on and recorded this as the RT (choosing from either the VPixx button release time, index finger, back of hand or medial wrist), with a minimum threshold of 40 frames (or 160 ms) to ensure a conscious response was captured (as opposed to non-physiological noise). This pipeline also allowed manually checking for any camera drop-out through the trials and ensuring key kinematics events were labelled and recorded correctly. The final step of the pipeline was to manually check and label each response into “Successful reach” (to each object in a Go trial), “No reach”, “Miss reach”, “Wrist movement” or “invalid trial” (removed due to recording error, which occurred when something occluded the objects’ ability to be tracked, such as the hand in an unusual position or accidentally knocking over the objects while trying to reach), as per the definitions above.

Once this pipeline was completed, the cleaned data were imported into jamovi Version 1.6, where all statistical analyses were performed [59]. Data were first examined for outliers using box plots and checked for violations of assumptions of normality with density plots and measures of skewness and kurtosis. Simple descriptive data were calculated for all demographic variables and all averaged task kinematic data. Paired *t*-tests were conducted for comparison of RT measurements, object location within each Go/No-Go task and to compare error response data with Go trial data. Individual error responses were examined separately in a post hoc exploration of varying error response behaviours.

## 3. Results

### 3.1. Demographic Data

The sample of *n* = 28 participants included 12 female and 16 male, aged 19–40 (*m* = 28.08, *sd* = 5.64). All the participants were right-hand-dominant, with a mean laterality quotient of 74.72 (*sd* = 23.04). The participants all scored normal to high non-verbal IQ on the Ravens Progressive Matrices Shortened version, with a mean of 48.93 (*sd* = 3.24). In normative studies, scores for this shortened version range from 30 to 48 [60], while the range for this sample was 39.79–54.86. A detailed breakdown of the autistic traits and depression, anxiety and stress are presented in Table 1 and Table 2, respectively.

To assess the participants’ eye–hand coordination, the SLURP test was administered with an iPad and iPencil, previously shown to be significantly faster and more accurate than using a rubber-tipped stylus [62] or a finger [63]. This included the full set of 20 shapes available in the app, allowing the shapes to be broken down into the categories reported in Table 3. The time taken to complete the SLURP is measured in seconds and averaged per shape in each category, while errors were defined as occurring when the stylus moved outside the shape; the average values per shape are reported.

A test of vigilant/sustained attention, arousal and performance throughout the trials was administered using a seven-item Likert scale following the completion of each reach and grasp inhibition task [64]. This allowed the participants to provide a subjective measure on how well they perceived their own attention to the task, i.e., whether concentration was maintained throughout the task and whether they believe they performed the tasks to the best of their ability. All the participants maintained moderate to high performance, as reported in Table 4.

Due to non-normality found in the majority of the data collected, non-parametric Kendall’s Tau B correlations were performed to assess the relationship between variables. Only significant correlations are reported here, with all the correlations reported in Appendix A. A strong significant correlation was found between anxiety and the AQ (*τb* (28) = 0.324, *p* = 0.025), and specifically the AQ5 Attention to Detail domain (*τb* (28) = 0.322, *p* = 0.030). A very strong, significant correlation was found between the total time taken to complete the SLURP with the AQ5 Attention to details (*τb* (22) = 0.381, *p* = 0.017). The time taken to complete the SLURP easy shapes was correlated with the AQ (*τb* (22) = 0.325, *p* = 0.039) and AQ5 Attention Switching (*τb* (22) = 0.327, *p* = 0.044), AQ5 Attention to Detail (*τb* (22) = 0.446, *p* = 0.005). The number of errors in the SLURP easy shapes was correlated with the DASS-21 (*τb* (22) = 0.415, *p* = 0.008) and stress (*τb* (22) = 0.383, *p* = 0.016). The time taken to complete the SLURP medium shapes was very strongly correlated with AQ5 Attention to Detail (*τb* (22) = 0.418, *p*= 0.009). The time taken to complete the SLURP hard shapes was significantly negatively correlated with the DASS-21 total score (*τb* (22) = −0.317, *p* = 0.042). The number of errors in SLURP hard shapes also correlated with DASS-21 total score (*τb* (22) = −0.317, *p* = 0.042) and the depression index (*τb* (22) = −0.429, *p* = 0.007).

### 3.2. Reaction Time Calculation

Motor RT was measured as described in Methods, allowing a comparison of RT in “Go” trials dependent on the measurement used (the traditional button release measured by VPixx and the kinematic measure of first marker to move as measured by Qualisys cameras). The breakdown of which marker moved first when calculating RT using the Qualisys method is illustrated in Figure 4, demonstrating that the camera-based back of hand and medial wrist markers tended to move prior to the index finger or button release. This provided a more consistent and accurate measure of RT, suggesting that, for rehabilitation exercises, videoing of hand and wrist markers is likely to be a more accurate and less costly RT measure than the electronic button press system.

The different methods used to define RT resulted in apparently shorter RT for the Qualisys-defined measure of RT than in the traditional button release measure of RT, as displayed in Figure 5.

The difference between the Qualisys-camera-defined RT (*m* = 239 ms, *SD* = 37) was significantly shorter than the electronically defined VPixx-defined RT (*m* = 324 ms, *SD* = 47), *t*(17) = 28.2, *p* < 0.001 (*Cohen’s d* = 6.66) for the middle practice task. The same was observed across all motor inhibition tasks: Left Go/No-Go, Qualisys (*m* = 332 ms, *SD* = 84) and VPixx (*m* = 409, *SD* = 89), *t*(17) = 18.2, *p* < 0.001 (*Cohen’s d* = 4.29); Middle Go/No-Go, Qualisys (*m* = 336 ms, *SD* = 65) and VPixx (*m* = 423 ms, *SD* = 94), *t*(17) = 13.4, *p* < 0.001 (*Cohen’s d* = 3.16) and Right Go/No-Go, Qualisys (*m* = 324 ms, *SD* = 80) and VPixx (*m* = 412 ms, *SD* = 91), *t*(17) = 17, *p* < 0.001 (*Cohen’s d* = 4).

### 3.3. Kinematic Data for Correct Go Trials

Detailed kinematic data for the middle control task can be found in Appendix A, and for the Go/No-Go tasks in Appendix A. The timeline of the kinematic events is presented in Figure 6.

### 3.4. Kinematic Data for No-Go Trials

There were three possible responses recorded during No-Go trials: the NR (no reach: for which no kinematic response data were recorded due to the participant remaining still), MR (miss reach: when the participant released the red button and made a partial movement towards the object, allowing a measure of RT and movement stop) and WM (wrist movement: where participants remained pressing the red button but the hand moved in response to the object illuminating, indicating a partial inhibition, also allowing a measure of RT and movement stop). Overall, the No-Go trials consisted primarily of NR trials (*n* = 889), with 8.4% identified as MR (*n* = 92) and 10.6% as WM (*n* = 116).

The full kinematic data are presented in Appendix A. The averaged trials collapsed across tasks are presented in Table 5, reporting the average RT for Go, WM and MR, and the average number of MR and WM recorded. The timelines of the error kinematics are also presented in Figure 6.

Three separate one-way nonparametric Kruskal–Wallis ANOVAs were conducted to compare variables of interest. First, comparing response type (reach to object, miss reach and wrist movement) revealed a significant difference for RT *X*^2^(4) = 256, *p* < 0.001 (*e*^2^ = 0.105) and time to stop movement *X*^2^(4) = 361, *p* < 0.001 (*e*^2^ = 0.150). When comparing task type (Go versus No-Go), there was also a significant difference for RT *X*^2^(1) = 233, *p* < 0.001 (*e*^2^ = 0.095) and time to stop movement *X*^2^(1) = 357, *p* < 0.001 (*e*^2^ = 0.148), although with a smaller effect size for both than when comparing response type. On the other hand, comparing target location revealed no significant results.

Non-parametric Kendall’s Tau B correlations were performed to assess the relationship between demographic variables and responses recorded during No-Go trials, with only significant correlations reported. The amount of WMs in the middle Go/No-Go task was significantly negatively correlated with the Raven’s average score (*τb* (24) = −0.337, *p* = 0.030). Overall, the average RT of WM trials was significantly negatively correlated with AQ5 Attention Switching (*τb* (24) = −0.389, *p* = 0.012), while the overall average RT for Go trials was significantly negatively correlated with the AQ5 Communication domain (*τb* (28) = −0.349, *p* = 0.021).

## 4. Discussion

The overall aim of this study was to develop better quantification of visuomotor behaviour and explore the sensitivity of variables impacting this in a group of least likely to be affected by stroke, young adults, as a means of refining understanding of potential measures of motor activity in populations impaired by neurodegenerative diseases and brain injury. As predicted, utilization of high-speed biological motion capture cameras allowed a more sensitive measure of the chronometry of behavioural responses in a motor Go/No-Go task. The first important finding was the large differences observed in measuring RTs based on movement rather than reliance on index finger button release. As shown in Figure 4, the back of hand and medial wrist were most often the first indicator of movement in response to the stimulus cue as opposed to the index finger and button release. This resulted in a significantly faster measure of response, by approximately 100 ms, which could result in more rigorously representative measurement of motor deficits [2] in addition to better elucidation of underlying neurological activity [18].

Comparison of our kinematic recording data with that of the original task by Castiello, Paulignan [47] indicates relative consistency. Our practice task involving only reaches to the middle object was similar to Castiello et al.’s results of simple reaches to the middle object, although we observed a shorter RT and longer movement discontinuation time (MT). However, when examining the Go trials in each of our Go/No-Go tasks, the RT to each object was much longer than those reported in the previous simple RtG task [47], suggesting that the added attention required to process the stimulus cue and make a response decision impacted the RT, consistent with Go/No-Go research [36]. While slight differences in kinematic profiles depending on the target object location were observed, these differences were not significantly different, suggesting that the differences between the responses were more advantageous to study than the simple motor behaviour observed in the Go trials.

The second crucial observation that this study allowed was in the examination of No-Go behaviour and analysis of error responses. The observation of “wrist movements” (WM) in response to the stimulus cue, whereby the participants continued pressing on the button and thereby did not produce an error but showed a movement of the wrist, indicating a “partial inhibition” [65], has major implications for both measuring underlying neural activity of error-related behaviour and also in measurement of motor deficits found in neurodegenerative and brain-injured patients. More WMs were observed than MRs in our data across all types of Go/No-Go tasks. While previous research has demonstrated non-smooth trajectories associated with error correction [11,12,13,14] and potentially with proactive control [15], to our knowledge, this is the first kinematic measure of this behaviour captured, and the first evidence allowing the timing of this behaviour to be isolated. Future research aligning this with neural activity has the potential to further our understanding of inhibition of motor behaviour and, more specifically, the errors made by stroke survivors.

The WM response observed was much later than the average Go or MR RT, although the responses showed large variability, indicating occurrence both early and late in the trial. This could reflect two distinct processing types, with an early WM reflecting an immediate inhibition of impulse [30] and a later WM reflecting a second guessing of the original inhibition and potentially a change in decision [66]. While MR kinematics were similar to Go trials, the added advantage of the kinematic recordings was the ability to measure the moment in time the movement was stopped, potentially capturing when the participant became conscious and activated the decision to cease making an error [67].

Further examination of the error responses revealed that MRs did not occur directly after a WM, suggesting that a WM potentially reinforces learning of inhibition [68] or enhances conscious attention [36]. Both WM and MR were more likely to occur directly after a completed reach during a Go trial, when a participant has potentially developed a pattern of reaching to the object. This was also reflected when examining time between trials, due to the nature of the task being self-paced and allowing participants to take short breaks between trials, and the number of previous reaches made before the error, due to the randomisation of the target presented, allowing continuous blocks of reaches before an inhibition target would appear. No differences were found between task type (or, therefore, inhibition target), but differences were observed between error response type. An MR would commonly occur after a longer sequence of Go trials, suggesting the participant had slowed down and potentially lapsed in attention. Both MR and WM commonly occurred following a longer period of consecutive reaches to Go targets, suggesting that the participant had become used to reaching and grasping and potentially forgot the rule to inhibit.

Our final aim was to explore demographic variables that could impact the behaviour on the task and provide relevant information regarding translation of these results to clinical populations. The study consists of an exploratory sample of neurotypical young adults with slightly higher than average non-verbal IQ [60] and all with strong right-hand preference [50]. On average, they scored lower than typically reported on the AQ [49], with higher scores relating to difficulties with attention on the five-subscale attention domains [61]. The sample had higher than average DASS-21 scores, but the percentiles reflected close to normal spread [48]. When comparing the Slurp Eye–Hand coordination results with normative data previously reported for participants using a rubber-tipped stylus [62], our sample appeared to take longer to complete the shapes, but this resulted in fewer errors than average.

Non-parametric correlations revealed a strong relationship between time taken to complete shapes on the Slurp and the AQ attention subscales for the AQ-5 [49]. These attention subscales refer to attention switching (higher scores indicating greater ability to focus intently on one aspect but difficulty switching attention between tasks) and attention to detail (higher scores indicating heightened awareness of fine details but a lack of awareness of the whole scene or bigger picture) [49]. This correlation of higher scores on these subscales with shorter RT on the Slurp task could suggest that the ability to note fine details in the shape, combined with a heightened focus on the task at hand, produced an ability to move less on the visuomotor tracing task. Interestingly, the number of errors made on the task was negatively correlated with the DASS-21, with fewer errors made by people who scored higher on stress/anxiety questions. This could suggest that traits of depression, anxiety and stress result in participants taking more care while completing a visuomotor task [69,70].

Similar to the findings with the Slurp task, higher scores on the AQ5 Attention Switching domain correlated with fewer WMs in No-Go trials, again suggesting that the hyper-focus element of the domain allowed the participants to maintain attention on the task at hand and remain more confident in their responses. This relationship was also observed with higher scores on the Raven’s Progressive Matrices, resulting in fewer WMs in No-Go trials in the Middle Go/No-Go task. Indeed, the number of WMs recorded in the No-Go trials was higher than MRs, which could reflect ongoing lapses of sustained attention [35,36] or working memory [71]. This is further evidence of high individual variability impacting inhibitory control [72] and emphasizes the need to examine trials individually both between and within subjects in stroke patients, who are more likely to experience fatigue and depression during rehabilitation [26,27,28].

Research into error responses suggests there is a “point of no return” during motor control where it is too late to inhibit [73] or correct [47] a particular motor response. Analysis of such motor behaviour confirms that correcting or stopping an erroneous response in a reaching movement requires approximately 260 ms [74] regardless of whether the finger, hand, wrist or arm is measured [75]. High individual variability can also impact analysis of both inhibitory control [72] and RT [76], particularly as individual RTs are not statistically independent and can vary trial to trial [77,78,79]. Such data highlight the probability that older persons post stroke are going to be even less able to inhibit incorrect movements and, hence, are likely to not reach, drop or fall once an incorrect movement is initiated by a lapse in attention.

An unfortunate limitation in this study, and in all ecological studies of error responses, is the small amount of error trials that occur. While Go/No-Go behaviour can be manipulated by the frequency of No-Go targets [34], it is difficult to manipulate the error rate, and, in order to generalize any behavioural findings, it would require extremely large sample sizes beyond the scope of the current study or most studies of older, healthy persons as compared to those post stroke. While the implications of these preliminary findings presented may not impact behavioural studies of the Go/No-Go task, they may have major implications for studies intending to characterize the underlying neural mechanisms. Future research intending to characterize the neural underpinnings with techniques that involve high temporal resolution would benefit from kinematic recordings of not only overt response behaviour but also characterization of potential stimulus-cued behaviour distinct from physiological noise, such as the WM we have described. While the sample in the current study is limited to young, healthy adults, this is applicable to comparison with young stroke survivors, who make up 5–30% of stroke patients, varying based on differing demographics [80], with the incidence rising in recent years [81], particularly in the context of the recent COVID-19 epidemic [82].

## 5. Conclusions

The use of high-speed biological motion capture cameras in this study allowed a more sensitive measure of the responses underlying visuomotor control. In combination with the variables collected on eye–hand coordination, mood, attention and intelligence, this measure holds potential to allow more accurate temporally sensitive measurements to be utilized in populations such as stroke patients, where visually driven motor behaviour is expected to be impaired and feelings of mortality vulnerability, anxiety, stress and depression are usually enhanced. Future studies in clinical populations utilizing this technology should provide much greater insight into the underlying neural processes relevant for motor rehabilitation.

## Figures and Tables

**Figure 1 brainsci-12-01581-f001:**
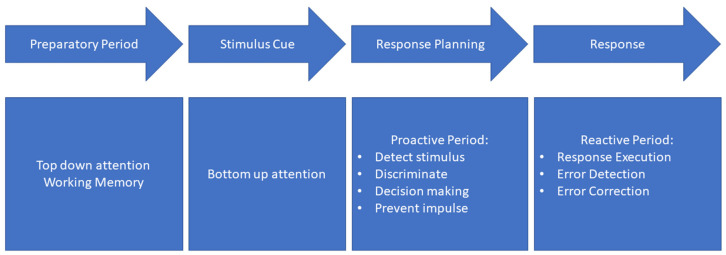
Go/No-Go underlying processes.

**Figure 2 brainsci-12-01581-f002:**
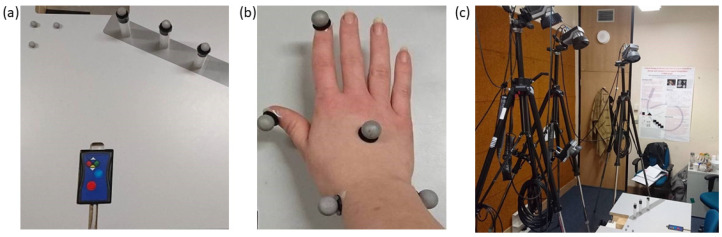
(**a**) Top-down view of the task apparatus: button box in centre of the participant’s midline with the blue button to indicate trial start and the red button to press and hold until objects illuminated one of the 3 cylindrical, translucent objects connected to the VPixx system (VPixx Technologies 2018) and programmed to illuminate, with reflective markers attached to the top for motion capture cameras. (**b**) A participant’s right hand with reflective markers attached to the index finger, thumb, back of hand, medial wrist and lateral wrist. (**c**) The set-up of the four Qualisys cameras [56] in relation to the task apparatus, consistent through all recordings.

**Figure 3 brainsci-12-01581-f003:**
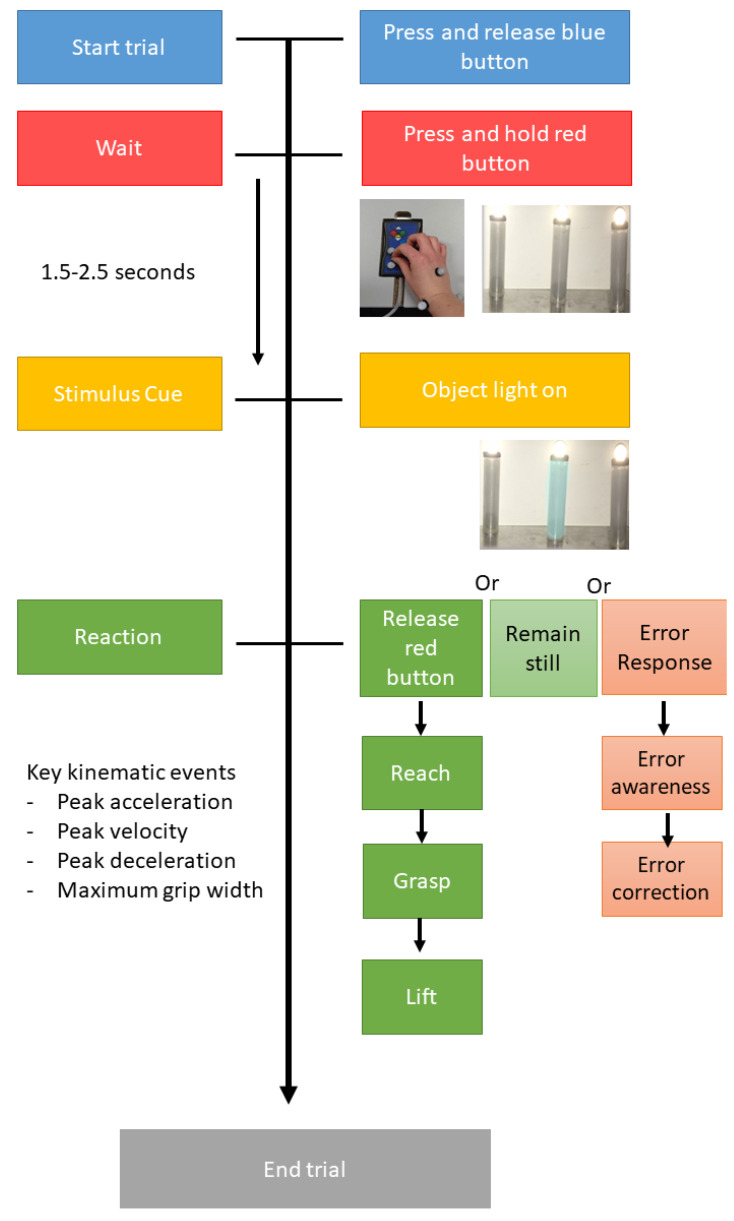
Timeline of task events and possible responses, from the start of the trial (blue), the attention period waiting for stimulus cue (red), the variable stimulus onset cue of light on (LO, yellow) and the reaction period (RP, green) during which a participant could reach and grasp, successfully inhibit or produce an error response. LO: light on; RT: reaction time; RP: reaction period; PA: peak acceleration; PV: peak velocity; PD: peak deceleration; MGW: maximum grip width.

**Figure 4 brainsci-12-01581-f004:**
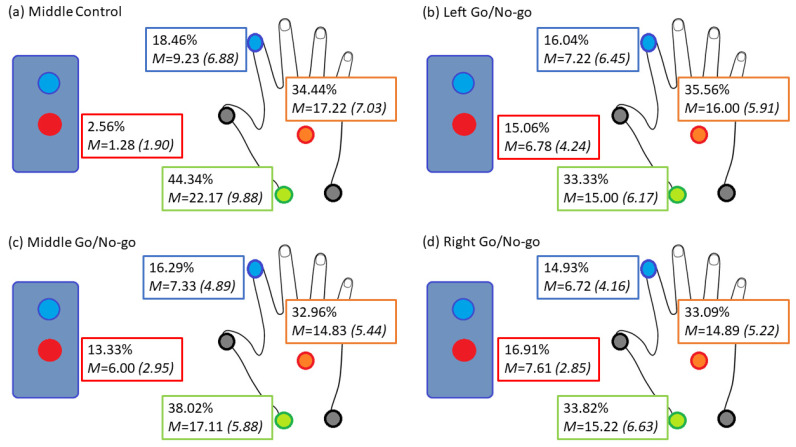
A schematic of the average percentage of how often each marker was used to calculate RT as the first measure of movement. Percentages are shown for each of the four tasks: for the middle practice block, left Go/No-Go, middle Go/No-Go and right Go/No-Go. Possible markers measured include the button release (red), index finger (blue), back of hand (orange) and medial wrist (green).

**Figure 5 brainsci-12-01581-f005:**
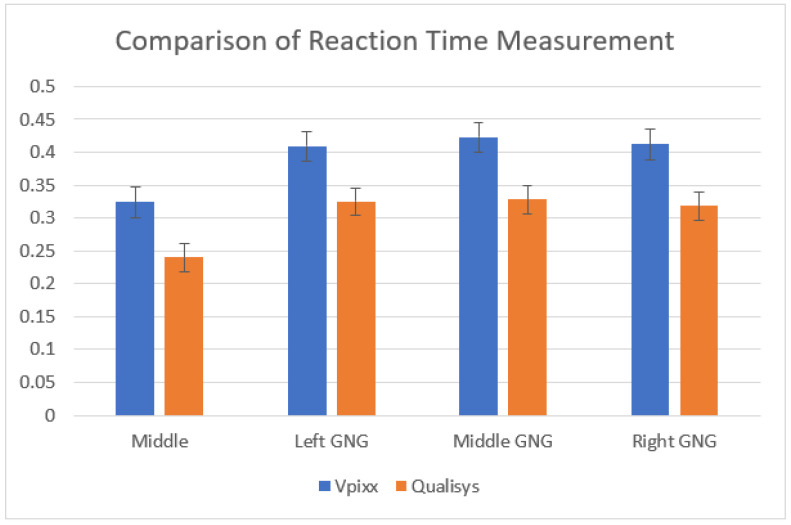
Means and standard errors of Qualisys- and VPixx-defined RTs for middle practice and each Go/No-Go task.

**Figure 6 brainsci-12-01581-f006:**
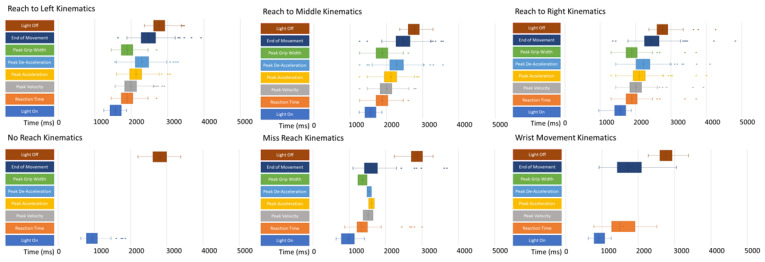
A box and whisker plot displaying the timelines of key kinematic variables averaged across trials, including mean, standard deviations and range of light on, reaction time, peak acceleration, peak velocity, peak deacceleration, maximum grip width, movement end and light off. Top row includes timeline for reach to left object, reach to middle object and reach to right object. Bottom row includes no reach, miss reach and wrist movement.

**Table 1 brainsci-12-01581-t001:** A detailed description of the Autism Quotient scores, which can be subcategorised into 5 domains as originally designed [49]. Normative values report from an Australian sample of AQ with 5 domains [61], adapted with permission from [61], Copyright © 2013 J. Broadbent et al.

	Study Sample (*n* = 28)		Normative Sample [61]	
Score Category (Range)	M	SD	M	SD
**AQ-Full with 5 Categories (7–25)**	15.1	4.80	21.74	14.56
Social Skills (0–8)	2.18	1.76	4.30	4.18
Attention Switching (1–7)	3.93	1.68	5.30	3.56
Attention to Detail (1–9)	4.89	2.38	5.04	2.31
Communication (0–5)	2.11	1.40	3.74	3.78
Imagination (0–5)	2.00	1.68	4.22	3.09

**Table 2 brainsci-12-01581-t002:** A detailed description of the DASS-21 with subscale scores for depression, anxiety and stress. Scores reported in raw form and converted into percentiles based on normative data [48].

	Raw Scores				Percentiles			
	M	SD	Min	Max	M	SD	Min	Max
**Total**	11.9	8.95	2	36	62	25.7	19	97
Depression	3.82	4.60	0	21	62.4	22.6	34	99
Anxiety	2.11	2.17	0	7	64.5	18.8	44	96
Stress	5.93	3.17	1	16	55.9	21.7	20	97

**Table 3 brainsci-12-01581-t003:** Detailed SLURP descriptive data (for *n* = 22 participants) for mean time taken to complete shape in seconds and mean number of errors made per shape for the total (*n* = 20), the easy (*n* = 7), medium (*n* = 9) and hard (*n* = 3) sets, in addition to the shortened Set A (*n* = 6) and Set B (*n* = 6), which both included 2 shapes from each difficulty category.

	Mean Time per Shape (S)				Mean Errors per Shape (*n*)			
	M	SD	Min	Max	M	SD	Min	Max
**Total**	34.1	14.3	17.9	87.4	4.22	1.83	2.26	8.74
Easy	23.6	11.7	13.9	59	3.08	2.90	0.57	8.43
Medium	36.8	16.8	18.3	103	3.85	1.94	1.44	9.89
Hard	50.4	19.4	24.3	119	7.98	3.98	3.33	18.7
**Set A**	30	13.9	15.1	83.2	2.89	2.04	0.67	7.33
**Set B**	31.6	12.7	16.3	79.6	3.02	1.60	0.33	7.33

**Table 4 brainsci-12-01581-t004:** Descriptive data for the 7-point Likert scale of self-reported levels of concentration, arousal and performance.

	M	SD	Min	Max
**Concentration**				
Left Go/No-Go	5.61	1.20	3	7
Middle Go/No-Go	5.71	1.30	3	7
Right Go/No-Go	5.57	1.43	2	7
**Arousal**				
Left Go/No-Go	5.71	1.24	2	7
Middle Go/No-Go	5.61	1.29	2	7
Right Go/No-Go	5.57	1.26	2	7
**Performance**				
Left Go/No-Go	5	1.05	3	7
Middle Go/No-Go	5.75	1.08	4	7
Right Go/No-Go	5.79	1.10	3	7

**Table 5 brainsci-12-01581-t005:** Overall averaged RT for Go, “No Reach” (NR), “Wrist Movement” (WM) and “Miss-Reach” (MR) with different task type collapsed; the number of WM and MR responses recorded on average.

	Reaction Time (ms)					Number of Response Type Recorded (*n*)				
	N	M	SD	Min	Max	N	M	SD	Min	Max
**Go**	28	320	76	219	535					
**WM**	24	697	385	94.7	1491	28	5.93	4.24	0	18
**MR**	24	309	187	81.8	932	28	3.14	2.41	0	8

## Data Availability

All data are available upon request.

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
