# Peer review of "Kinematic Studies of the Go/No-Go Task as a Dynamic Sensorimotor Inhibition Task for Assessment of Motor and Executive Function in Stroke Patients: An Exploratory Study in a Neurotypical Sample"

_brainsci, 2022, doi:10.3390/brainsci12111581_

Round 1

Reviewer 1 Report

Varied motion capture techniques have been widely used in investigating the reach-to-grasp movements among individuals with neurological disorders. It is anticipated that the high-speed cameras provide more/better information than the button box (as shown in Fig. 5). The reaction time of the button box could be even higher if it had a longer system delay. Therefore, the result in Fig. 5 is unsurprising. In addition, the authors should discuss if the differences in Fig 5 are a systematic bias. If yes, the differences would not be a big deal.  

Please reconsider the data analysis. If my understanding is correct, there are three within-subject factors:

Task type (Go trials (baseline session), Go/No-Go trials)

Response type (Correct Reach, No Reach, Miss reach, Wrist Movement)

Target position (left, mid, right)

I think the mean RT values of each subject (in each condition) should be fitted to a repeated-measures ANOVA. Reading the results from the paired t-tests in the Result session is confusing.

Reach-to-Grasp is a widely used protocol in behavioral studies, so I don’t think the authors need to introduce it in detail (Lines 64-96). Meanwhile, the authors should explicitly show us the novelty finding in the Result session and tells us why this experiment can expand our knowledge of Neurobehavior. 

Author Response

We firstly thank the reviewers for their insightful comments and believe their feedback has greatly improved the manuscript. Please see attached.

Reviewer 2 Report

Dear Editor,

in the manuscript entitled “Kinematic studies of the “Go” and “No-Go” task as a dynamic sensorimotor inhibition task for assessment of motor and executive function in stroke patients: An exploratory study in a neurotypical sample”, the authors explored 28 healthy participants’ kinematic profiles of hand movements utilizing a Go/NoGo task with the aim to refine the temporal trajectory of the kinematic measurement captured using time-sensitive motion capture cameras.  Kinematics profiles are evaluated to improve the assessment of sensorimotor function impairment in post-stroke patients.

This is a well-conceived and interesting study. However, in my view the manuscript is not ready.

These are my comments:

1.     Overall, I believe the authors have explored common parameters of kinematics and much more can be done. In particular, I believe that the study would be enriched by evaluating other measures reported in the literature, for instance parameters related to the presence or absence of corrections to the motor plane: one-shot or non-one-shot movements. More in details, the one-shot movements reflect a regular movement without alteration of the motor command (Flash & Hogan, 1985; Morasso, 1981), while the non-one-shot movements are characterized by a non-linear / multipeaked trajectory, with one or more corrections and indicate an alteration with respect to the initial motor plane (Fishbach et al., 2005, 2007, Milner & Ijaz, 1990; Novak et al., 2002). To get a better idea, I suggest reading the article by Benedetti et al. (2020). I know the movements measured in this study are different, but if this analysis can be done I think it could be very informative about evaluating visuomotor resources in stroke survivors. Especially for what concerns the evaluation of creation/implementation of the motor plan.

References

Benedetti, V.; Gavazzi, G.; Giovannelli, F.; Bravi, R.; Giganti, F.; Minciacchi, D.; Mascalchi, M.; Cincotta, M.; Viggiano, M.P. Mouse Tracking to Explore Motor Inhibition Processes in Go/No-Go and Stop Signal Tasks. Brain Sci. 202010, 464. https://doi.org/10.3390/brainsci10070464

 Flash, T., and Hogan, N. (1985). The coordination of arm movements: an experimentally confirmed mathematical model. J. Neurosci. 5, 1688–1703. doi: 10.1523/JNEUROSCI.05-07-01688.1985

 Fishbach, A., Roy, S. A., Bastianen, C., Miller, L. E., and Houk, J. C. (2005). Kinematic properties of on-line error corrections in the monkey. Exp. Brain Res. 164, 442–457. doi: 10.1007/s00221-005-2264-3

 Fishbach, A., Roy, S. A., Bastianen, C., Miller, L. E., and Houk, J. C. (2007). Deciding when and how to correct a movement: discrete submovements as a decision making process. Exp. Brain Res. 177, 45–63. doi: 10.1007/s00221-006-0652-y 

 Milner, T. E., and Ijaz, M. M. (1990). The effect of accuracy constraints on three-dimensional movement kinematics. Neuroscience 35, 365-374. doi: 10.1016/0306-4522(90)90090-q 

 Morasso, P. (1981). Spatial control of arm movements. Exp. Brain Res. 42, 223–227. doi: 10.1007/BF00236911

 Novak, K., Miller, L., and Houk, J. (2002). The use of overlapping submovements in the control of rapid hand movements. Exp. Brain Res. 144, 351-364. doi: 10.1007/s00221-002-1060-6 

2.      In methods. Authors describe the protocol: “Participants were first instructed to start each trial, when ready, by pressing the blue button closest to them on the button box. They were instructed to then immediately press and hold down the red button with their index finger and thumb in a pincer position, and wait for the cue to reach out and pick up the object as quickly and as accurately as possible when a dowel was illuminated. On trial completion, the dowel was returned to its original position and the blue button was to be pressed to start the next trial.” Despite some citations reported below, I have found the task design described in a very complex way. In my opinion this section is not clearly explained. To be precise, it is understandable, but the reading cannot be so hard in a journal with the audience of Brain Sciences. Can the authors explain the task paradigm more clearly in the text?

3.      In methods. Authors report:”manually check and label each response into “Successful reach” (to each object in a Go trial), “No reach”, “Miss reach”, “Wrist movement” or “invalid trial” (removed due to recording error), as per the definitions above.”. What does it mean removed due to recording error; can the authors be more specific?

4.      In the discussion section, the authors report the limitations of the study. I believe that there is a big limitation in this study not reported. That is, the age of the sample enrolled (“The sample of n=28 participants included 12 female and 16 male, aged 19-40 - m=28.08, 265 sd=5.64”) is too young if compared to the typical age of people having a stroke. I think authors should add this limitation to the discussion.

Author Response

(The authors gave the same response as above.)

Round 2

Reviewer 1 Report

All my comments have been addressed. Best wishes for the authors future study. 

Reviewer 2 Report

The paper has improved enough to be published.